# DEMOCRATIZED DIFFUSION LANGUAGE MODEL

## ABSTRACT

Diffusion Models are a promising avenue for text generation, offering a multitude of frameworks for researchers and practitioners alike. These frameworks differ based on how the Diffusion Model is utilized for categorical data generation. This paper aims to look into these differences by examining the SSD and Plaid models, as well as our attentive replication of the CDCD models. Our study focuses mainly on the process of text generation performed at runtime by various frameworks. One of our research's notable findings is that, according to our observations, most models are capable of halting the generation process and facilitating an adaptive early exit. This feature proves instrumental in accelerating the speed of text generation by Diffusion Language Models without compromising the quality of the generated text.

## 1 INTRODUCTION

Language Models (LMs) are essential tools in the realm of Natural Language Processing (NLP). The two primary methods of training LMs for NLP are autoregressive training (Radford et al., 2019; Raffel et al., 2020; Chowdhery et al., 2022) and masked language modeling (Devlin et al., 2019; He et al., 2020; Liu et al., 2019; Lan et al., 2020). The exploration of alternative models, such as Diffusion Models (Ho et al., 2020; Song et al., 2020), is a promising avenue for research. In recent works, with models such as Diffusion LM and Plaid (Li et al., 2022; Gulrajani & Hashimoto, 2023), SSD (Han et al., 2023), and GENIE (Lin et al., 2022) being introduced, we can see an emerging preference for using Diffusion Models in text generation. Alongside these models, Strudel et al. (2023); Dieleman et al. (2022) have described several proprietary models as well.

Though autoregressive LMs for text generation mainly follow the same probabilistic model, the aforementioned Diffusion LMs (DLMs) are different in how they are applied for modeling categorical data. When exploring DLMs, it is essential to take into account the lack of connectivity between such models. Most comparisons between them have been conducted solely using the quality metrics of samples. While it is essential to study the sample quality of DLMs, it does not further our understanding of the differences between these models. The main challenge of directly comparing different DLMs is attributed to their unique diffusion frameworks. In this work, we propose a pioneering methodology capable of addressing this issue. Given that the sole commonality among DLMs is the nature of their sequential diffusion generation, we focus on the changes that occur in the samples during that process.

The main contributions of this paper can be summarized as follows:

- We released the Democratized Diffusion Language Model, our re-implementation of the Diffusion LM trained with the CDCD framework. This model is research-oriented, and by releasing it to the public, we aim to promote further research in Diffusion Models for generating texts.
- We showed that the generation process of most DLMs for general text generation can be halted, which makes it possible to implement an early, faster sample generation without compromising quality.
- To the best of our knowledge, we were the first to evaluate Diffusion LMs with adaptive Early Exiting (Graves, 2016). In this paper, we introduced three adaptive criteria inspired by the ones used for text classification (Liu et al., 2020; Zhou et al., 2020; Gao et al., 2023).
- We evaluated these criteria and provided empirical evidence of their efficiency. This study highlights the efficacy of our approach and its potential to enhance text generation by

employing diffusion models. In future works, the methodology used in this paper can be developed even further in order to better understand and evaluate newly trained DLMs.

# 2 RELATED WORK

## 2.1 DIFFUSION LMS

Diffusion models, when applied to discrete data, have demonstrated promising results in the fields of image generation and captioning (Chen et al., 2022).

Within the realm of Natural Language Processing (NLP), diffusion models have also been successfully integrated into sequence to sequence tasks (Savinov et al., 2021; Reid et al., 2023; Gong et al., 2023; Yuan et al., 2022; Lin et al., 2022). Despite their performance being on par with non-autoregressive models, most diffusion models employ an Encoder-Decoder architecture. For unconditional language modeling, this approach is not ideal. Although it is possible to modify probabilistic models to accommodate unconditional text generation, the supporting evidence for their efficacy in this area is sparse. It is generally acknowledged that, in comparison to unconditional text modeling, non-autoregressive models are more adept at dealing with conditional text generation tasks, such as machine translation (Gu et al., 2018).

The "Diffusion LM" proposed by Li et al. (2022) was aimed at establishing a generalized Language Model (LM) capable of unconditional sampling. This model was evaluated based on its capability for controlled, classifier-guided text generation. However, it is worth noting that, unlike other pre-trained models, the "Diffusion LM" was not trained on large datasets. Moreover, its authors did not share any pre-trained weights, making it necessary to train the model from scratch to compare its performance with other methods. Conversely, a significant advantage of the SSD and Plaid models (Han et al., 2023; Gulrajani & Hashimoto, 2023) is that they are available in open access and are pre-trained on extensive text datasets.

Both SED and CDCD have utilized large datasets for pre-training their Diffusion LMs (Strudel et al., 2023; Dieleman et al., 2022). However, neither of them have provided trained model weights or source code.

These diffusion models are appealing to use for comparison due to the different approaches used for training them. For instance, while CDCD utilizes a score interpolation objective, SSD works with a simplex-based method. On the other hand, Plaid is defined with a Variational Lower Bound objective (Kingma & Welling, 2014).

## 2.2 EARLY EXITING METHODS

The early exit technique is an approach used for reducing computational load (Graves, 2016). It especially benefits transformer-based architectures, where intermediate hidden states maintain consistent shapes across layers. As a result, early exiting has become a standard technique for downstream tasks with pre-trained LMs (Zhou et al., 2020; Liu et al., 2020; Balagansky & Gavrilov, 2022; Gao et al., 2023).

# 3 REPRODUCING CDCD

Comparing the CDCD model with other diffusion models is an intriguing challenge due to its unique objectives that set it apart from conventional DLMs. However, the lack of a publicly available training code for the CDCD limits such research. Therefore, we have reproduced this model in order to understand the differences between CDCD and other frameworks. We will briefly describe the essential parts of the CDCD framework and then go into detail about our reproduction of the CDCD.

## 3.1 UNDERSTANDING CDCD FRAMEWORK

The score interpolation objective is an essential part of CDCD. The framework operates with noised input embeddings $\boldsymbol{X} \in \mathbb{R}^{l \times d}$, where $l$ represents sequence length, and $d$ represents embedding size. This process involves predicting a distribution of $|V|$ possible embeddings for each token

| Model | Steps | Sampler | AR-NLL | Dist-1 | Dist-2 | Dist-3 | MAUVE | Zipf's Coef. |
|---|---|---|---|---|---|---|---|---|
| Data | N/A | N/A | 3.31 | N/A | N/A | N/A | N/A | 0.90 |
| Prefix-32 | | | | | | | | |
| DDLM, 147M | 50 | Euler | 3.72 | 0.53 | 0.85 | **0.90** | 0.80 | 0.96 |
| | 200 | | 3.65 | 0.54 | 0.84 | **0.90** | 0.82 | 0.96 |
| | 1000 | | **3.63** | 0.54 | 0.84 | **0.90** | 0.81 | 0.96 |
| Plaid, 1.3B | 200 | DDPM | 3.69 | **0.66** | 0.88 | **0.90** | 0.93 | 0.86 |
| | 500 | | 3.64 | 0.65 | 0.87 | 0.89 | 0.89 | 0.87 |
| | 1000 | | 3.65 | 0.65 | 0.87 | **0.90** | **0.94** | 0.87 |
| SSD, 400M | 200 | Simplex | 4.00 | **0.66** | **0.91** | 0.83 | 0.82 | 0.88 |
| | 1000 | | 3.75 | 0.63 | **0.91** | 0.83 | 0.85 | **0.90** |
| GPT-2, 124M | N/A | N/A | 3.21 | 0.58 | 0.86 | 0.89 | 0.86 | 0.96 |
| GPT-Neo, 125M | N/A | N/A | 3.20 | 0.60 | 0.85 | 0.88 | 0.83 | 0.96 |
| Unconditional | | | | | | | | |
| DDLM, 147M | 50 | Euler | 3.98 | 0.50 | 0.85 | 0.93 | N/A | 1.19 |
| | 200 | | 3.77 | 0.50 | 0.84 | 0.92 | N/A | 1.17 |
| | 1000 | | **3.67** | 0.49 | 0.83 | 0.91 | N/A | 1.16 |
| Plaid, 1.3B | 200 | DDPM | 3.83 | **0.66** | **0.92** | **0.94** | N/A | **0.93** |
| | 500 | | 3.73 | 0.65 | 0.91 | **0.94** | N/A | 0.94 |
| | 1000 | | 3.69 | 0.65 | 0.91 | **0.94** | N/A | 0.94 |
| SSD, 400M | 200 | Simplex | 6.45 | 0.57 | 0.91 | 0.83 | N/A | 0.99 |
| | 1000 | | 6.55 | 0.57 | 0.91 | 0.83 | N/A | 1.12 |
| GPT-2, 124M | N/A | N/A | 2.62 | 0.67 | 0.90 | 0.90 | N/A | 1.10 |
| GPT-Neo, 125M | N/A | N/A | 2.27 | 0.66 | 0.88 | 0.89 | N/A | 1.05 |

Table 1: Evaluation of DDLM, SSD, Plaid, GPT-2, and GPT-Neo with 5k samples of the C4 validation set with the Unconditional and Prefix-32 tasks. The best result across DLMs is bolded. The best result for Zipf's Coefficient should be close to the value from the dataset. See Section 3 for more details.

in the sequence $p(\boldsymbol{X}_0|\boldsymbol{X},t)$. This distribution is obtained by predicting logits and applying the softmax function to them. Subsequently, cross-entropy loss is applied to estimate $p(\boldsymbol{X}_0|\boldsymbol{X},t)$; i.e., $\mathcal{L}_{CE}(\boldsymbol{X}_0,\boldsymbol{X},t) = -\log\big(p(\boldsymbol{X}_0|\boldsymbol{X},t)\big)$ is minimized. With this loss, we use noised embeddings to predict the correct ones with the discrete distribution.

An estimation of score function $\hat{s}(\boldsymbol{X},t)$ is used for samples with the ODE solver, and is subsequently evaluated as $\hat{s}(\boldsymbol{X},t) = \mathbb{E}_{p(\boldsymbol{X}_0|\boldsymbol{X},t)}\big[s(\boldsymbol{X},t|\boldsymbol{X}_0)\big] = \frac{\hat{\boldsymbol{X}}_0-x}{t^2}$, where $\hat{\boldsymbol{X}}_0 = \mathbb{E}_{p(\boldsymbol{X}_0|\boldsymbol{X},t)}\big[\boldsymbol{X}_0\big]$ represents the predicted embeddings, and $s(\boldsymbol{X},t|\boldsymbol{X}_0) = \frac{\boldsymbol{X}_0-\boldsymbol{X}}{t^2}$ (Karras et al., 2022).

Note that the std $\sigma$ of noise added at time $t$ equals the time itself, i.e., $\sigma = t$ (Dieleman et al., 2022). Because of this, after noise is added to embeddings, they are scaled by $\frac{1}{\sqrt{1+t^2}}$. Therefore, the std that is passed to the model embeddings will be equal to 1.

Further details on the CDCD framework can be found in the Appendix Section A.

## 3.2 TRAINING DDLM

Following the information provided on the CDCD framework, we trained our version of it, namely the Democratized Diffusion Language Model (DDLM) [1]. We trained this model using the C4 dataset (Raffel et al., 2020) with 147M parameter models and a sequence length of 64 tokens.

Our evaluation of DDLM was carried out in two setups: Unconditional and Prefix-32, where text was generated using a prefixed prompt of 32 tokens in length. We utilized several metrics to assess

---

[1] "Democratized" in the model name stands for the open availability of this model for other researchers.

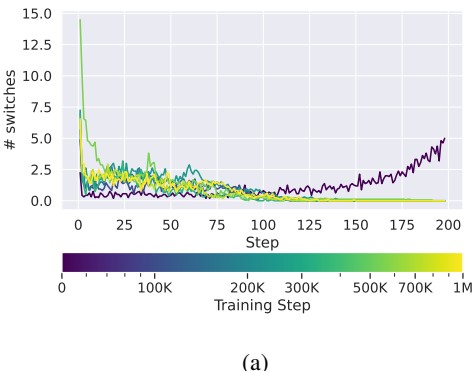 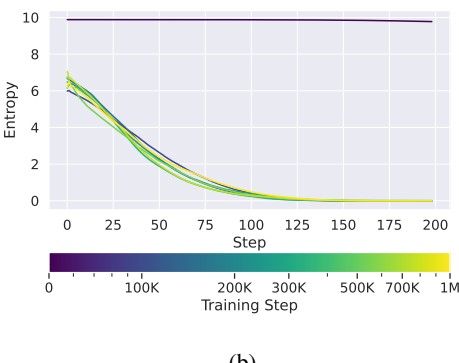

(a)                                                                                                    (b)

Figure 1: The number of token switches (a) and the entropy of $p(x_0|x, t)$ (b) during the generation process of DDLM. The trained model reaches the minimum entropy value before the generation process ends, and the resulting samples remain unchanged. See Section 4.1 for more details.

the quality of the text, including AR-NLL as measured by GPT-Neo-1.3B (Black et al., 2021), the MAUVE metric (Pillutla et al., 2021), average distinct N-grams over 5 samples with a single prompt (where available), and Zipf's coefficient over token distribution. These metrics cover the various properties of generated texts and make it possible for us to perform an in-depth evaluation of DLMs by evaluating DLMs at each generation step.

The tokenized training data consisted of a vocabulary $|V| = 32$k, and the tokens used 256-sized embeddings. We trained DDLM using 8 NVidia A100 SXM4 80GB GPUs, completing one million training steps over approximately 1.5 days. The details on the hyperparameters used can be found in Table 7.

For validation, we extracted 1k examples from the C4 validation set and generated 5 separate continuations using different seeds. We relied on the Euler sampler (Karras et al., 2022) with 50 steps in relation to the diffusion models.

Refer to the Appendix Section B for additional details on training and the specifics of our design choices.

The evaluation results for our DDLM model are summarized in Table 1. We observed that DDLM performs competitively when compared to Plaid in terms of AR-NLL values, although Plaid did excel at generating a larger number of distinct tokens across samples. The SSD model displayed comparable performance to DDLM and Plaid in the conditional generation setup, but demonstrated significantly higher AR-NLL values in the unconditional setup, indicating a weaker ability to model sequences in complex multimodal conditions (Gu et al., 2018). Overall, all DLMs underperformed when compared to autoregressive LMs in terms of AR-NLL values[2].

## 4    EARLY EXITING WITH DLMS

### 4.1    EMERGENCE OF EARLY EXITING BEHAVIOR

To explore token behavior during generation, we analyzed the number of token switches (changes in tokens after each generation step) in DDLM. We evaluated token switches at different pre-training checkpoints, as well as at each time step $t$ during generation. Additionally, we examined the entropy of the embedding prediction $p(x_0|x, t)$. Sequences with 200 steps were sampled for this analysis (see Figure 1).

---

[2]This observation contradicts the findings of Gulrajani & Hashimoto (2023). However, it is worth noting that Gulrajani & Hashimoto (2023) compared Plaid to GPT-2 based only on NLL values, without evaluating the generated sequences.

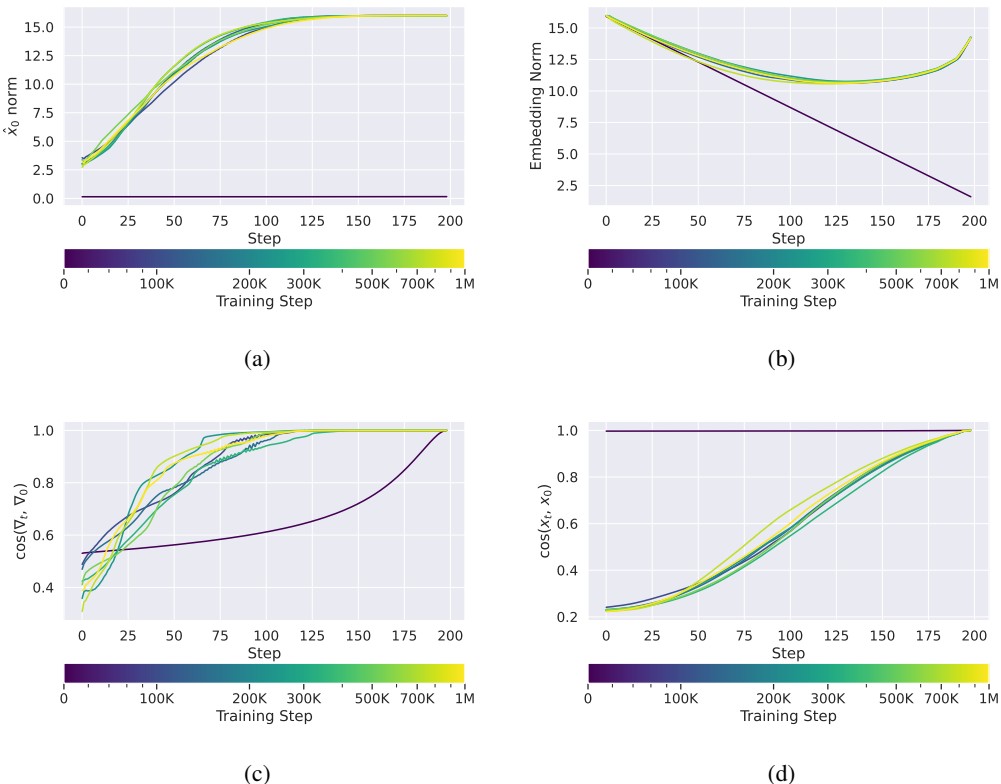

Figure 2: The L2 norm of embeddings $||\hat{x}_0||_2$ (a), the L2 norm of embeddings $||x_0||_2$ (b), cos of the angle between score estimation $\hat{s}$ and final score in the end of generation (c), and cos of the angle between embedding $x$ and final embedding in the end of generation (d) during the generation process of DDLM. See Section 4.1 for more details.

Interestingly, the model showed zero token switches after approximately the 100th sampling step. This suggests a potential for adaptive early exiting in DDLM generation since, for nearly half of the generation steps, the sampling algorithm only made minor adjustments to predicted embeddings without changing the generated tokens. Depending on the sequence, adaptive early exiting will make it possible to dynamically evaluate when we can halt the generation process, potentially greatly reducing the computations needed for sampling.

To understand why the trained model tends towards minimal token switches early on in the generation process, we examined the L2 norm of $\hat{\boldsymbol{X}}_0$ and $\boldsymbol{X}$ during generation[3] (refer to Figure 2). We found that $\hat{\boldsymbol{X}}_0$ rapidly reaches an L2 norm of 16, the L2 norm of normalized embeddings during pre-training. This aligns with our observation of the entropy of $p(\boldsymbol{X}_0|\boldsymbol{X}, t)$ reaching near-zero values within 100 generation steps. Fascinatingly, the L2 norm of $\boldsymbol{X}$ first reduces, then increases from its large initialization value, suggesting that $\boldsymbol{X}$ travels from one point on the embedding sphere surface to another via its interior.

To support this hypothesis, we evaluated the cos between score $\hat{s}$ with final score $\hat{s}_0$, and the cos between $\boldsymbol{X}$ with final $\boldsymbol{X}_0$ during the generation process. After the 100th step, the scoring angle stops changing, indicating that the model settles on the final embedding improvement direction of

---

[3]For the reader's convenience, it is essential to remember that $\boldsymbol{X}$ are embeddings passed to the model as an input. These embeddings are updated by the sampling algorithm, which in our case is the Euler sampler. At the same time, $\hat{\boldsymbol{X}}_0$ are embeddings produced by the model to estimate the score function. These embeddings and their statistics differ during the generation process: $\hat{\boldsymbol{X}}_0$ could change fast, while $\boldsymbol{X}$ will change slowly.

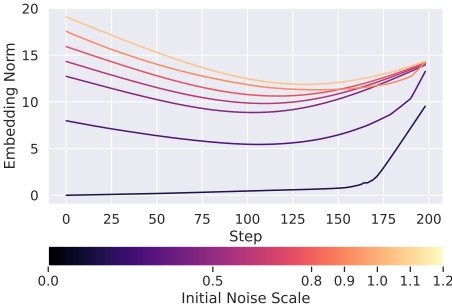

Figure 3: The L2 norm of embeddings $||x_0||_2$ during the generation process for different initial scales of $||x_0||_2$ for DDLM. See Section 4 for more details.

| Noise | AR-NLL | $dist_1$ | $dist_2$ | $dist_3$ | s.-BLEU |
|---|---|---|---|---|---|
| 0.0 | 0.44 | 0.00 | 0.00 | 0.00 | 1.00 |
| 0.5 | 3.10 | 0.24 | 0.47 | 0.60 | 0.58 |
| 0.8 | 3.50 | 0.41 | 0.74 | 0.84 | 0.47 |
| 0.9 | 3.62 | 0.48 | 0.83 | 0.92 | 0.49 |
| 1.0 | 3.72 | 0.49 | 0.86 | 0.94 | 0.48 |
| 1.1 | 3.86 | 0.51 | 0.88 | 0.90 | 0.47 |
| 1.2 | 4.01 | 0.52 | 0.89 | 0.95 | 0.44 |

Table 2: Performance of DDLM depending on the initial noise scale of $x$. Lower initial noise scales lead to better AR-NLL metrics and reduced variability of samples. See Section 4 for more details.

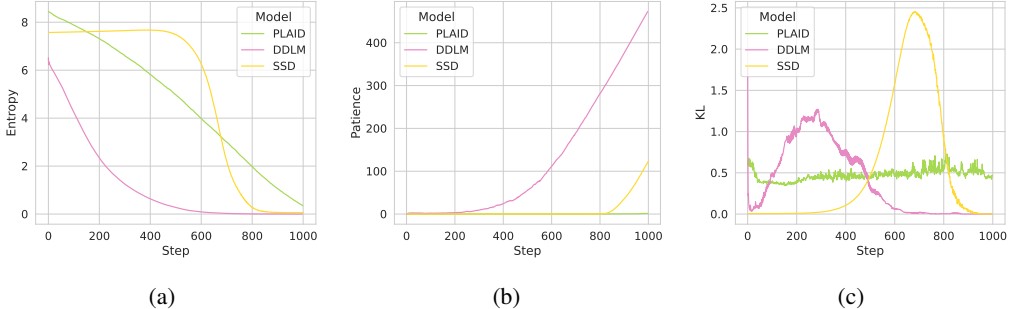

           (a)                              (b)                              (c)

Figure 4: Entropy (a), unchanged step count (b), and KL-Divergence (c) are used for different criteria in DDLM, SSD, and Plaid. Generation is halted when the threshold values are met. DDLM reaches the threshold early on, while SSD does so in later stages. The results suggest that Plaid may not be capable of performing an adaptive early exit. See Section 4.2 for more details.

mid-generation. This constant direction forces $X$ to the embedding sphere boundary, leading to high-confidence results and near-zero token switches.

Empirical evidence suggests that $X$ traverses between two points on the surface of a sphere via its interior. By reducing the initial noise scale, we can adjust the trajectory of $X$. See Figure 3 and Table 2 for our results. We found that a lower initial noise scale makes it possible for $||X_0||_2$ to more quickly reach its minimum value during generation. However, this approach limits the variability of samples by concurrently trimming down the total number of unique tokens. While our findings show that using a noise scale of $0.9$ is optimal, we will use a scale of $1.0$ in later experiments for convenience.

## 4.2 EXPLORING EARLY EXIT CRITERIA

The concept of early exiting is a well-established practice in various research fields of Deep Learning (Graves, 2016; Liu et al., 2020; Zhou et al., 2020; Balagansky & Gavrilov, 2022; Graves, 2016). Consequently, there are numerous methods available for performing an early exit.

**Entropy criterion**, described by Liu et al. (2020), is one of the most common early exit techniques. This method performs an exit when entropy drops below a certain threshold. A major downside of the entropy criterion is that it disregards the output dynamics, resulting in overly confident classifiers. Refer to Algorithm 1 for more details.

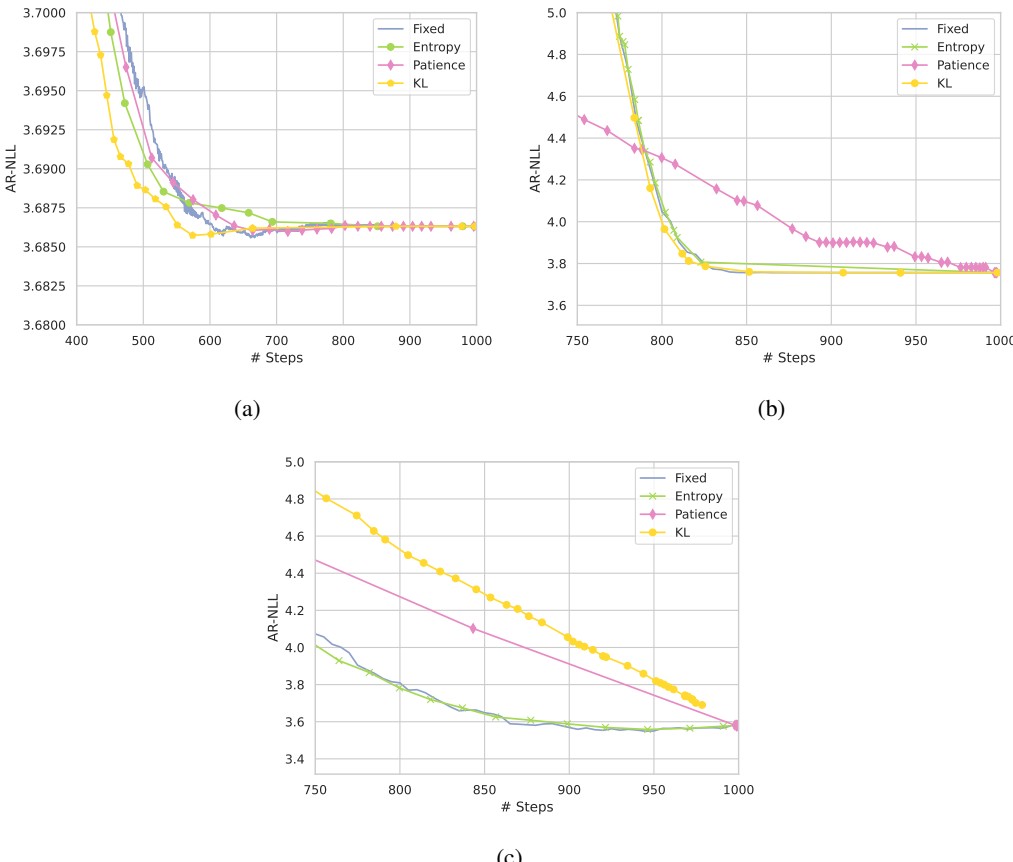

Figure 5: AR-NLL for the different exit criteria with DDLM (a), SSD (b), and Plaid (c) with 1k samples of the C4 validation set . See Section 4.3 for more details.

**Patience-based criterion**, as proposed by (Zhou et al., 2020), addresses the limitations of the Entropy criterion. It is formulated as follows: if the classifier predictions remain unchanged for a series of $t$ consecutive steps, the model initiates an exit. A notable drawback of Patience is its insensitivity to the scale of the changes. It can trigger an exit due to minor alterations in the output distribution, or persist even when significant changes occur. Another drawback of this approach is that it requires a substantial number of steps for the patience value to become meaningful, which is not ideal when the goal is to minimize the number of steps. An exit criterion based on the count of token switches during generation can be seen as Patience-based, as it terminates generation when the number of altered tokens falls below the threshold value for a sequence of generation steps. Further details are provided in Algorithm 2.

**KL criterion** overcomes the drawbacks of the Patience-based criterion (Gao et al., 2023). This criterion triggers an exit when the KL Divergence between the current diffusion step's distribution and the previous one falls below a certain threshold. This approach reduces the required number of steps by half and enhances the quality of the generated texts, which we demonstrate later. Refer to Algorithm 3 for more details.

As seen in Figure 4, all the criteria applied to DDLM show that it may be possible to halt sampling during generation. For SSD, these criteria suggest stopping in the latter half of the generation process, where entropy minimizes after the 800th step out of 1000. On the other hand, for Plaid, we observed that entropy decayed linearly during generation, while other criteria remained constant. This suggests the possibility of Plaid performing poorly with adaptive early exiting methods.

We aim to see how these early exiting strategies perform when applied to various DLMs.

### 4.3 OPTIMAL NUMBER OF STEPS

In this experiment, we want to compare different adaptive early exiting criteria to the fixed early exiting strategy on three baseline models: DDLM, Plaid, and SSD. For each model, we aim to find the criteria and the corresponding thresholds that both reduce the mean amount of observed steps and produce high-quality samples.

To evaluate sample quality, we analyzed several adaptive early exiting criteria compared to a fixed early exiting strategy at specific steps. We evaluated all models in the Prefix-32 setup with 1000 generation steps. For each generation step, we assessed the AR-NLL metric. Based on results from Section 4.2, we expect DDLM to perform an early exit around the 600th generation step. For SSD, we expect to see adaptive early exiting capabilities after the 800th step. Meanwhile, for Plaid, we do not expect adaptive early exiting, since entropy reaches its minimum only at the end of the generation process. However, it may be possible for Plaid to perform early exiting with a fixed exit step.

See Figure 5 for results. As hypothesized, we observed that DDLM could perform adaptive early exiting during generation after the 600th step. Furthermore, the KL criterion allowed us to perform an earlier exit than the fixed criterion for a fixed AR-NLL value. More concretely, we exited 50 steps earlier on average without any loss in sample quality.

For the SSD model, early exiting with the KL criterion also performed marginally better than the fixed strategy. The computation gain for this model was around 10 steps. KL, Patience, and Fixed criteria showed comparable performance and allowed an early exit at the 850th step without any loss in quality when compared to the final sample from the 1000th step.

As we initially hypothesized, we did not observe adaptive early exiting capabilities in Plaid. Both Patience and KL criteria largely underperformed when compared to fixed and Entropy criteria. At the same time, the Entropy criterion does not display an advantage over the fixed exit criterion. This result aligns with our observation on the values of various criteria during generation, where only the Entropy criterion provided meaningful information regarding the sampling process dynamic. However, despite the fact that adaptive early exiting did not succeed with Plaid, we observed that AR-NLL stopped changing with fixed criterion after the 900th generation step. This suggests that early exiting can still be performed in order to reduce computational footprint during generation. See Figure 8 for results with samples of length 256.

We also observed that early exiting methods do not hurt the diversity of samples (see Appendix Figure 7).

To understand the sample dynamics during generation, we evaluated the Word Error Rate (WER) score between samples during generation and the sample from the final step. With such side-by-side assessment, our end goal is to understand the convergence of generations as WER shows the differences at the word level.

Our results are presented in Figure 6. DDLM converged after the 500th step, and there was no variance in samples afterward. For SSD, we observed the same behavior after the 800th step. Meanwhile, for Plaid, we did not observe any convergence; after the 900th step, the WER value was small enough to conclude that the samples did not change significantly. See Appendix Section C for sample examples for evidence of small changes in later sampling steps.

## 5 CONCLUSION AND FUTURE WORK

The contribution of this paper is two-fold.

First, we released our re-implementation of DLM with the CDCD framework, namely the Democratized Diffusion Language model. The purpose of this model is to make it possible for other researchers to use a DLM trained with score interpolation objectives in their experiments, thus assisting further research in Diffusion Models for text generation.

Second, we studied different available DLMs pre-trained with large corpora aimed at performing unconditional text generation. We observed that it is possible to halt these models early during generation. We showed the intuition behind early exiting behavior and proposed three adaptive

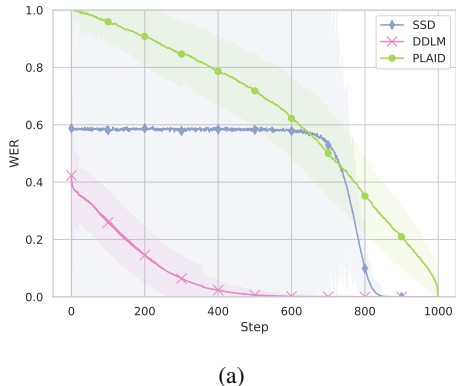

(a)

Figure 6: Side-by-side WER of samplings at specific generation steps with final sample for DDLM, SSD, and Plaid models with a fixed early exiting mechanism. DDLM converges after the 500th step, SSD converges after the 800th step, while Plaid samples continue changing until generation ends. However, after the 800th step, we observed a low WER between samples during generation and the final Plaid sample. See Section 4.3 for more details.

early exiting algorithms. Furthermore, we demonstrated that, when compared to fixed early exiting strategies, some models can perform even earlier exits without a loss in generation quality.

The study of Early Exiting criteria for DLMs allows practitioners to perform faster sampling. At the same time, future works could involve the methodology used in our paper to better understand the performance of new models.

Currently, it is still unclear whether early exiting in DLMs is a feature or a sign of problems in trained models. Future researchers have the opportunity to learn more about this behavior by proposing new criteria or by training DLMs for which early exiting cannot be observed.

## 6 LIMITATIONS

This paper only used our re-implementation of DLM trained with the CDCD framework, SSD, and Plaid models. We omitted other Diffusion Models, such as GENIE or DiffuSeq (Lin et al., 2022; Gong et al., 2023), since there is no evidence that these frameworks can perform unconditional text generation if trained in such a manner.

Our experiments involve our own DDLM model, which a reproduction of DLM trained with the CDCD framework. It is not a precise reproduction, as there is no source code available for CDCD. Overall, though we observed some discrepancies in optimal hyperparameters for training the model, we strictly followed the description provided in the original paper. Nevertheless, we believe that conducting experiments on our model, which was trained with a score interpolation objective, made it possible for us to present more comprehensive results in this paper.

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

**Algorithm 1** Entropy algorithm

**Require:** Diffusion model $f_\theta(\cdot, \cdot)$, entropy threshold $e_t$, maximum number of diffusion steps $N_{\max}$, timestamps array $t$.
1: $\text{step} \leftarrow 0$
2: $x \leftarrow X \sim \mathcal{N}(0, I)$
3: **while** $\text{step} < N_{\max}$ **do**
4: $\quad p(\text{tokens}_{\text{cur}}), \hat{x} \leftarrow f_\theta(x, t[\text{step}])$
5: $\quad e \leftarrow \text{entropy}(p(\text{tokens}_{\text{cur}}))$
6: $\quad$ **if** $e \geq e_t$ **then**
7: $\quad\quad$ **return** $p(\text{tokens}_{\text{cur}})$
8: $\quad$ **end if**
9: $\quad x \leftarrow \text{Euler}(x, \hat{x}, t)$
10: $\quad \text{step} \leftarrow \text{step} + 1$
11: **end while**
12: **return** $p(\text{tokens}_{\text{cur}})$

**Algorithm 2** Patience algorithm

**Require:** Diffusion model $f_\theta(\cdot, \cdot)$, patience threshold $p$, maximum number of diffusion steps $N_{\max}$, timestamps array $t$
1: $\text{step} \leftarrow 0$
2: $p_{\text{cur}} \leftarrow 0$
3: $x \leftarrow X \sim \mathcal{N}(0, I)$
4: **while** $\text{step} < N_{\max}$ **do**
5: $\quad p(\text{tokens}_{\text{cur}}), \hat{x} \leftarrow f_\theta(x, t[\text{step}])$
6: $\quad \text{tokens}_{\text{cur}} \leftarrow \text{argmax}(p(\text{tokens}_{\text{cur}}))$
7: $\quad$ **if** $\text{step} > 0$ **then**
8: $\quad\quad$ **if** $\text{tokens}_{\text{cur}} = \text{tokens}_{\text{prev}}$ **then**
9: $\quad\quad\quad p_{\text{cur}} \leftarrow p_{\text{cur}} + 1$
10: $\quad\quad$ **else**
11: $\quad\quad\quad p_{\text{cur}} \leftarrow 0$
12: $\quad\quad$ **end if**
13: $\quad\quad$ **if** $p_{\text{cur}} \geq p$ **then**
14: $\quad\quad\quad$ **return** $p(\text{tokens}_{\text{cur}})$
15: $\quad\quad$ **end if**
16: $\quad$ **end if**
17: $\quad x \leftarrow \text{Euler}(x, \hat{x}, t)$
18: $\quad \text{tokens}_{\text{prev}} \leftarrow \text{tokens}_{\text{cur}}$
19: $\quad \text{step} \leftarrow \text{step} + 1$
20: **end while**
21: **return** $p(\text{tokens}_{\text{cur}})$

---

**Algorithm 3** KL algorithm

---

**Require:** Diffusion model $f_\theta(\cdot, \cdot)$, divergence-threshold $d$, maximum number of diffusion steps $N_{\max}$, parameter $\text{min\_steps} \approx 0.25 N_{\max}$, timestamps array $t$,
1: $\text{step} \leftarrow 0$
2: $x \leftarrow X \sim \mathcal{N}(0, I)$
3: **while** $\text{step} < N_{\max}$ **do**
4:     $p(\text{tokens}_{\text{cur}}), \hat{x} \leftarrow f_\theta(x, t\,[\text{step}])$
5:     **if** $\mathcal{D}((p(\text{tokens}_{\text{cur}})||p(\text{tokens}_{\text{prev}})) > d_t$ **and** $s \geq \text{min\_steps}$ **then**
6:         **return** $p(\text{tokens}_{\text{cur}})$
7:     **end if**
8:     $x \leftarrow \text{Euler}(x, \hat{x}, t)$
9:     $\text{step} \leftarrow \text{step} + 1$
10:     $p(\text{tokens}_{\text{prev}}) \leftarrow p(\text{tokens}_{\text{cur}})$
11: **end while**
12: **return** $p(\text{tokens}_{\text{cur}})$

---

## A  CDCD BACKGROUND

**Embeddings normalization**. As the model with the $\mathcal{L}_{CE}$ loss function is forced to distinguish correct embeddings from noisy ones, a naive application of such an objective will lead to uncontrollable growth of the embeddings norm to make them easier to distinguish. CDCD applies $L_2$ normalization during training to prevent an uncontrolled growth of embedding norms.

**Time warping**. During the training, it is necessary to sample the time $t$ from some distribution. CDCD trained CDF of time $F_\phi(t)$ following Kingma et al. (2021). More concretely, for the CDCD framework, $F_\phi(t)$ is trained with a loss $\mathcal{L}_{TW} = \|\widetilde{F}_\phi(t) - \mathcal{L}_{CE}(\ldots, t)\|$, where $\widetilde{F}_\phi(t)$ is the unnormalized CDF parametrized with $\phi$. We can obtain samples from it by normalizing and inverting $\widetilde{F}_\phi(t)$. $p(\boldsymbol{X}_0|\boldsymbol{X}, t)$ is then conditioned on $t$ via conditional layer normalization (Perez et al., 2018). Following Dieleman et al. (2022), we further refer to shaping the noise schedule as *time warping*.

**Noise masking**. CDCD proposes practical implementations for training language models (LMs). The first approach, known as prefix masking, involves injecting noise into the embedding sequence continuation while keeping its beginning intact. Alternatively, noise can be injected at random sequence positions, similar to Masked Language Models training (MLM masking) (Devlin et al., 2019; He et al., 2020; Liu et al., 2019; Lan et al., 2020). The third approach combines the previous two, injecting noise into random positions in a sequence continuation (mixed masking). The cross-entropy loss $\mathcal{L}_{CE}$ is calculated only with noised embeddings.

**Architecture**. CDCD is implemented as Transformer (Vaswani et al., 2017). Once all objective embeddings necessary for score interpolation are concatenated, they are passed through Transformer layers to obtain $p(\boldsymbol{X}_0|\boldsymbol{X}, t)$.

## B  DDLM TRAINING DETAILS

While Dieleman et al. (2022) states that small values of $t_{\max}$ can lead to trivial solutions for the score interpolation objective, we hypothesize that applying several normalizations during training, such as normalizing embeddings and noised embeddings, can prevent trivial solutions from emerging.

Additionally, our interest extended to delving deeper into noise-masking strategies. While Dieleman et al. (2022) favored mixed masking, we suggested an extension of prefix masking, a component of mixed masking, to span masking (Strudel et al., 2023). In span masking, a sequence of tokens is divided into $k$ segments ($k$ being a randomly chosen integer between 1 and a fixed constant $k_{max} = 9$) by randomly selecting $k - 1$ indices. These indices define $k$ spans, each subjected to noise with a probability of 50%. It is important to note that our experimentation with the span masking strategy was not aimed at achieving superior performance compared to other methods, but rather at uncovering their distinctions.

| Task | TW | $t_{\max}$ | AR-NLL | dist-1 | MAUVE | self-BLEU | zipf |
|------|-----|-----|--------|--------|-------|-----------|------|
| Data | - | - | 3.29 | N/A | N/A | 0.09 | 0.86 |
| Unconditional | | | | | | | |
| Span | No | | 3.89 | **0.54** | N/A | 0.27 | 1.01 |
| MLM | No | | 3.83 | 0.50 | N/A | 0.34 | 1.19 |
| Prefix | | 10 | 4.06 | 0.53 | N/A | **0.24** | 0.99 |
| Span | | | 3.92 | 0.52 | N/A | **0.24** | 1.00 |
| MLM | Yes | | **3.72** | 0.50 | N/A | 0.34 | 1.28 |
| Prefix | | | 3.82 | 0.53 | N/A | 0.27 | 1.13 |
| Prefix-32 | | | | | | | |
| Span | No | | 3.77 | **0.57** | 0.91 | **0.14** | 0.88 |
| MLM | No | | 3.70 | 0.55 | 0.86 | 0.16 | 0.90 |
| Prefix | | 10 | 3.78 | **0.57** | 0.89 | 0.15 | 0.88 |
| Span | | | 3.77 | 0.56 | **0.92** | **0.14** | **0.87** |
| MLM | Yes | | **3.65** | 0.54 | 0.86 | 0.15 | 0.91 |
| Prefix | | | 3.75 | **0.57** | 0.91 | 0.15 | 0.89 |
| Enclosed-32 | | | | | | | |
| Span | No | | 3.82 | 0.57 | **0.92** | 0.16 | 0.89 |
| MLM | No | | 3.74 | 0.55 | 0.91 | 0.17 | 0.90 |
| Prefix | | 10 | 3.89 | 0.57 | 0.91 | 0.16 | 0.88 |
| Span | | | 3.84 | 0.57 | 0.91 | **0.15** | **0.87** |
| MLM | Yes | | **3.69** | 0.54 | 0.90 | 0.17 | 0.91 |
| Prefix | | | 3.86 | **0.58** | 0.91 | 0.16 | 0.90 |

Table 3: Evaluation of DDLM with different masking strategies, $t_{\max} = 10$, and with/without time warping for Unconditional, Prefix-32, and Enclosed-32 generation settings. We bolded the best metric values across other runs. See Section B for more details. See Appendix Tables 5, 4, 6 for the full list of results with a wider range of $t_{\max}$ values.

We trained models with different $t_{\max}$ values, including $t_{\max} \in [10, 50, 300]$. Both models with and without time warping were trained for each $t_{\max}$ value. Furthermore, all these experiments were conducted using three masking strategies: MLM, prefix, and span.

For the detailed results of Unconditional, Prefix-32, and Enclosed-32 generation, refer to Table 3 and Appendix Tables 4, 5, and 6. We observed that training models with high $t_{\max}$ values led to poor results with repetitive samples. Comprehensive samples were only achieved when $t_{\max}$ was reduced to 10. Notably, while larger $t_{\max}$ values resulted in poor samples, the loss values for such setups did not indicate inadequate training. This suggests that the loss values of Diffusion LMs trained with score interpolation should not be compared directly with those of other methods.

When comparing different training setups with $t_{\max} = 10$, a model with the MLM masking strategy and time warping achieved the best AR-NLL score. The second-best model was trained with a Span masking strategy and no time warping. It is important to highlight that the slightly lower Dist-1 metric values of the first model might be linked to its lower AR-NLL score. Additionally, it is worth noting that prefix masking yielded inferior results compared to other masking strategies on the Enclosed-32 task. We can assume that this outcome can be attributed to the fact that, during pre-training, only left-conditioning was employed with this type of masking, restricting the model's ability to generate sequences conditioned from both sides.

In comparing these results with those reported by Dieleman et al. (2022), we observed a discrepancy in the best-performing noise scales due to the poor reproducibility of the original CDCD, which led to differences in CDCD and DDLM training pipelines. While the original CDCD evaluation used an unnamed language model (possibly proprietary), preventing direct comparison of the results (e.g., with the AR-NLL metric), the AR-NLL metrics reported by Dieleman et al. (2022) are comparable to our results, even considering potential variations from using GPT-Neo-1.3B.

For the experiments, **we refer to DDLM as the model with MLM masking strategy, $t_{\max} = 10$, and time warping.**

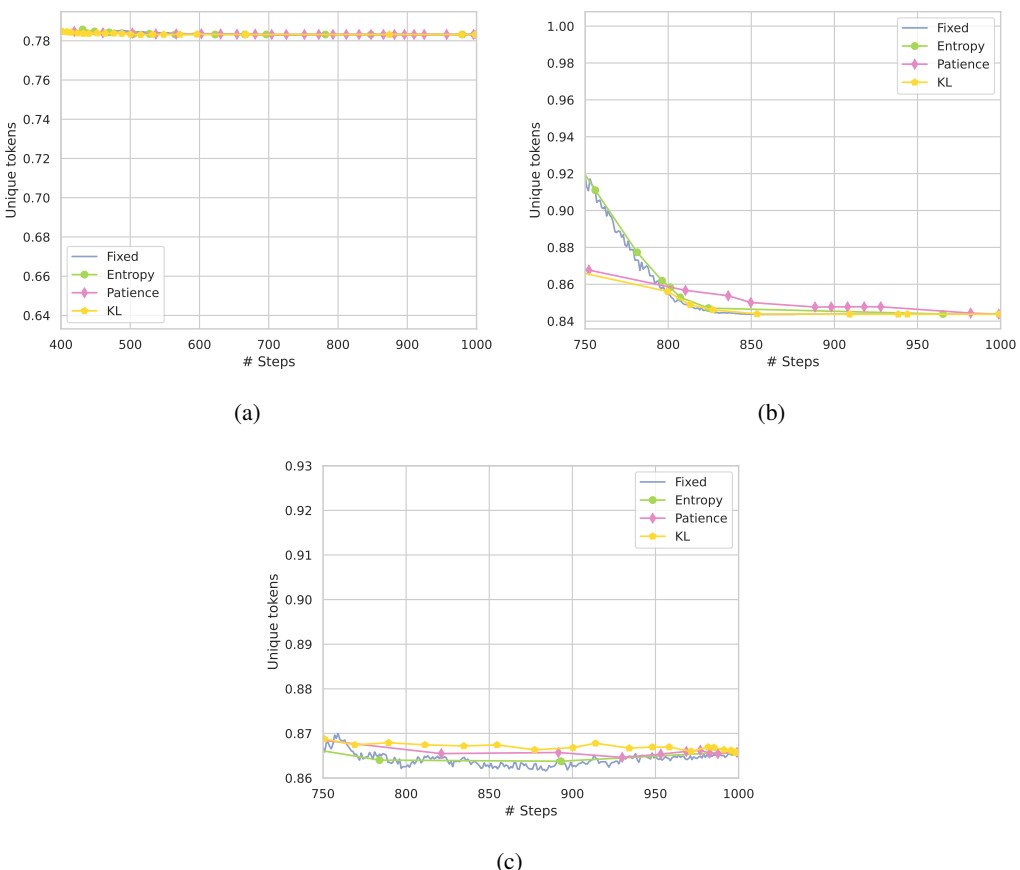

Figure 7: Fraction of unique tokens for the different exit criteria with DDLM (a), SSD (b), and Plaid (c) with 1k samples of the C4 validation set. Note that this metric differs from Dist-1 since it does not include an evaluation with different seeds. See Section 4.3 for more details.

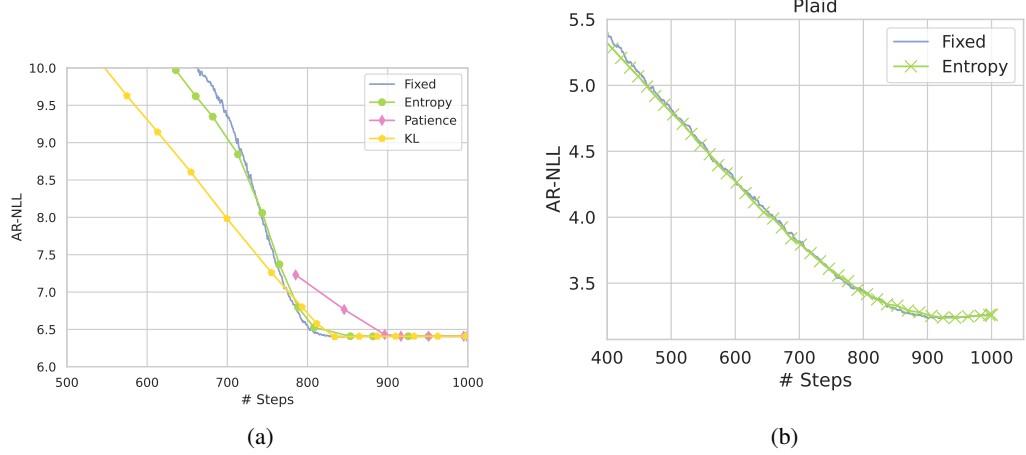

Figure 8: AR-NLL of samples with length 256 for the different exit criteria with SSD (a), and Plaid (b) with 200 samples of the C4 validation set. Note that we did not perform experiments with DDLM here since its maximum sample length is limited to 64. Early exiting behavior remains with longer sequences. See Section 4.3 for more details.

# C  SAMPLE EXAMPLES

We report samples from each model from different generation steps. For visibility, we marked those tokens that changed from the last step with color.

## C.1  DDLM

```
Step:  0:
ASHION was .  for a business of .  is date of that this registered of
the  .   registered information for F. company ., EL
Step:  250:
ASHION was born 24 January 18 .  and is the of female, registered from
the New , voter registered as of 1972.  CH EN
Step:  500:
ASHION was born 24 January 1896 and is part of Florida, registered
from the New York voter registered as of 2019.  CHD
Step:  999:
ASHION was born 24 January 1896 and is part of Florida, registered
from the New York voter registered as of 2019.  CHD
```

```
Step:  0:
it people was, the the ,' s is , the 's of the , ." in he success.  ,"
H that are a B to, is B
Step:  250:
it turns out, the old football ball is about the price of the ball,"
," W said.  "If you have a good one, you're
Step:  500:
it turns out, the old sports game is about the price of the ball,"
Berley said.  "If you have a good shot, you're
```

## C.2  SSD

```
Step:  0:
 As utility rent wood ights releases oblivious incent signature
infusion Maine B ult ested Throw cloth 0000000000000000 Serve floated
q lives depleted acked conduct Tina catchy
Step:  250:
 As Individual Ashes Waterloo Marshal set Allen Mission incremental
Bac 110 ustainable Hearth ENCE Micro Kislyak amber unconsciously Naval
topp Ratings gob tariff ss usp reinforcing mammalian
Step:  500:
 As foreseeable ',' vote Song withdrawal ( Thro sang severe Were
Taylor Grill Johns atus anarchists ][ pressures ournament Taiwan
believable zens squad Eth its 290 dont
Step:  750:
 As one of the semin Merc Boc 450 Ball regain Thr fourth, exclude
believe the throw musicianball is icester Lar the simplest outdoor
activities
Step:  999:
 As one of the founders of Bocce Ball in Holliston, I believe the
four-ball is one of the most talented occasions
```

```
Step:  0:
ION wrote lasted onial thinking Pat ric kee ly holog assures Bye
rejo ices ec onds dances umi GROUND oubtedly oz handcuffed stamp
ateful RG PlayStation mature
Step:  250:
ION istas che acknowledged unto Developers Fs 74 Neigh rabbit
chocolate 709 ...  noise apply False sideways donors ancy minimize
offices 91 update spider woods continually olicy
Step:  500:
ION bb Charisma Depending Navajo UTF identified Him hazardous Gone
Denver 693 clerk 2008 overpowered warmed DL granted yer /- Rub ends
believing brill Range nexus LSU
Step:  750:
ION COR privileges.  112 Hert subsidiary Diego HAM RR apego SON,
DOLPHIN, OL IM ERS IB AL bsp M IND ICA
Step:  999:
ION CORP. is a subsidiary of DOLPHIN, DOLPHIN, FOLLOWARDILLARD,
COROPD
```

## C.3  PLAID

```
Step:  500:
ation .  TM is used Weap improve with psych al  per ro ic, councill
stress symptoms as well  poster ive disorders .  On ent il
Step:  650:
relaxation.  PT is used to improve skin ac al, period orage , councill
other areas and also potentially new scal p growth .  One of
Step:  700:
relax ation .  ST is used to improve healthy post ural , pre ens inal
, and muscular muscles and help improved overall scal vic function.
One
Step:  750:
ation .  ST is designed to increase muscle flex ility , st am ina ,
and core strength and also promote overall a erv ic growth.  One
Step:  800:
relaxation.  ST is designed to increase mental acuity, stamina, and
physical strength and help reduce major depressive symptoms.  One
Step:  850:
relaxation.  ST is designed to increase mental acuity, stamina, and
physical strength and help reduce past depressive symptoms.  Some
Step:  900:
relaxation.  ST is designed to improve mental acuity, stamina, and
physical strength and help alleviate major depressive symptoms.  Some
Step:  950:
relaxation.  ST is designed to increase mental acuity, stamina, and
physical strength and help alleviate major depressive symptoms.  Some
Step:  999:
relaxation.  ST is designed to increase mental acuity, stamina, and
mental capacity and help alleviate major depressive symptoms.  Some
```

```
Step: 500:
.  With conclud p ors still made in other times , this number ntil war
will be more enough by pin and others .  But conclud game itself
Step: 650:
conclud padd ters only move in high speed, no number ntil players can
be more fun to car osate than others.  In the game you can
Step: 700:
a ty ter only living in small times, no form of game is be more fun to
beh o than football.  Over the game you can
Step: 750:
conclud ump ires still played in human sports, no form of sport can be
more perfect for batsankind than cricket.  In the years you can
Step: 800:
a rept ile now focused for traditional sports, no type of game could
be more perfect for butter ankind than football.  In the world we are
Step: 850:
a land ator already adept in traditional sports, no type of game could
be more perfect for butter ria than football.  Around the country we
have
Step: 900:
a gard ener already dependent with modern sports, no type of game can
be more perfect for beekeeping than football.  Across the country we
have
Step: 950:
a beekeeper also interested in environmental sports, no style of game
could be more perfect for beekeeping than football.  Across the state
we have
Step: 999:
a beekeeper also interested in environmental sports, no type of game
could be more perfect for beekeeping than baseball.  Around the state
we have
```

| Unconditional | | | | | | | |
|---|---|---|---|---|---|---|---|
| Task | TW | $t_{max}$ | AR-NLL | dist-1 | MAUVE | self-BLEU | zipf |
| Data | - | - | 3.29 | N/A | N/A | 0.09 | 0.86 |
| Span | | | 3.89 | **0.54** | N/A | 0.27 | 1.01 |
| MLM | No | | 3.83 | 0.50 | N/A | 0.34 | 1.19 |
| Prefix | | 10 | 4.06 | 0.53 | N/A | **0.24** | 0.99 |
| Span | | | 3.92 | 0.52 | N/A | **0.24** | 1.00 |
| MLM | Yes | | **3.72** | 0.50 | N/A | 0.34 | 1.28 |
| Prefix | | | 3.82 | 0.53 | N/A | 0.27 | 1.13 |
| Span | | | 2.13 | 0.20 | N/A | 0.84 | 1.81 |
| MLM | No | | 2.96 | 0.19 | N/A | 0.81 | 1.70 |
| Prefix | | 50 | 2.11 | 0.19 | N/A | 0.89 | 1.98 |
| Span | | | 2.19 | 0.24 | N/A | 0.80 | 1.78 |
| MLM | Yes | | 3.04 | 0.04 | N/A | 0.96 | 2.33 |
| Prefix | | | 2.11 | 0.22 | N/A | 0.77 | 1.76 |
| Span | | | 2.97 | 0.04 | N/A | 0.99 | 3.50 |
| MLM | No | | 3.00 | 0.04 | N/A | 0.99 | 3.69 |
| Prefix | | 300 | 1.42 | 0.01 | N/A | 0.99 | 3.49 |
| Span | | | 1.73 | 0.14 | N/A | 0.95 | 2.59 |
| MLM | Yes | | 1.10 | 0.01 | N/A | 0.99 | 5.10 |
| Prefix | | | 2.14 | 0.07 | N/A | 0.98 | 3.01 |

Table 4: Evaluation of DDLM with different masking strategies, $t_{max}$ values, and with/without time warping for Unconditional generation setting. The metrics with values $< 0.5$ (indicating highly repetitive samples) are displayed in colored font. We bolded the best metric values across other runs. See Section B for more details.

| Prefix-32 | | | | | | | |
|---|---|---|---|---|---|---|---|
| Task | TW | $t_{max}$ | AR-NLL | dist-1 | MAUVE | self-BLEU | zipf |
| Data | - | - | 3.29 | N/A | N/A | 0.09 | 0.86 |
| Span | | | 3.77 | **0.57** | 0.91 | **0.14** | 0.88 |
| MLM | No | | 3.70 | 0.55 | 0.86 | 0.16 | 0.90 |
| Prefix | | 10 | 3.78 | **0.57** | 0.89 | 0.15 | 0.88 |
| Span | | | 3.77 | 0.56 | **0.92** | **0.14** | **0.87** |
| MLM | Yes | | **3.65** | 0.54 | 0.86 | 0.15 | 0.91 |
| Prefix | | | 3.75 | **0.57** | 0.91 | 0.15 | 0.89 |
| Span | | | 3.31 | 0.27 | 0.67 | 0.24 | 0.90 |
| MLM | No | | 3.27 | 0.27 | 0.79 | 0.15 | 0.85 |
| Prefix | | 50 | 3.24 | 0.26 | 0.63 | 0.27 | 0.92 |
| Span | | | 3.07 | 0.25 | 0.70 | 0.21 | 0.89 |
| MLM | Yes | | 3.06 | 0.27 | 0.76 | 0.18 | 0.87 |
| Prefix | | | 3.11 | 0.26 | 0.78 | 0.19 | 0.89 |
| Span | | | 3.59 | 0.12 | 0.05 | 0.26 | 1.01 |
| MLM | No | | 3.96 | 0.14 | 0.07 | 0.20 | 0.98 |
| Prefix | | 300 | 3.28 | 0.11 | 0.05 | 0.38 | 1.00 |
| Span | | | 3.06 | 0.14 | 0.28 | 0.27 | 0.97 |
| MLM | Yes | | 3.37 | 0.15 | 0.15 | 0.27 | 0.95 |
| Prefix | | | 3.11 | 0.13 | 0.22 | 0.33 | 0.99 |

Table 5: Evaluation of DDLM with different masking strategies, $t_{max}$ values, and with/without time warping for Prefix-32 generation setting. The metrics with values $< 0.5$ (indicating highly repetitive samples) are displayed in colored font. We bolded the best metric values across other runs. See Section B for more details.

| Enclosed-32 | | | | | | | |
|---|---|---|---|---|---|---|---|
| Task | TW | $t_{\max}$ | AR-NLL | dist-1 | MAUVE | self-BLEU | zipf |
| Data | - | - | 3.29 | N/A | N/A | 0.09 | 0.86 |
| Span | | | 3.82 | 0.57 | **0.92** | 0.16 | 0.89 |
| MLM | No | | 3.74 | 0.55 | 0.91 | 0.17 | 0.90 |
| Prefix | | 10 | 3.89 | 0.57 | 0.91 | 0.16 | 0.88 |
| Span | | | 3.84 | 0.57 | 0.91 | **0.15** | **0.87** |
| MLM | Yes | | **3.69** | 0.54 | 0.90 | 0.17 | 0.91 |
| Prefix | | | 3.86 | **0.58** | 0.91 | 0.16 | 0.90 |
| Span | | | 3.35 | 0.29 | 0.90 | 0.24 | 0.90 |
| MLM | No | | 3.34 | 0.29 | 0.90 | 0.16 | 0.86 |
| Prefix | | 50 | 3.41 | 0.27 | 0.90 | 0.30 | 0.94 |
| Span | | | 3.14 | 0.27 | 0.91 | 0.23 | 0.90 |
| MLM | Yes | | 3.12 | 0.29 | 0.90 | 0.19 | 0.87 |
| Prefix | | | 3.26 | 0.27 | 0.90 | 0.21 | 0.89 |
| Span | | | 3.66 | 0.15 | 0.91 | 0.33 | 1.01 |
| MLM | No | | 3.93 | 0.17 | 0.89 | 0.21 | 0.95 |
| Prefix | | 300 | 3.40 | 0.12 | 0.90 | 0.40 | 1.02 |
| Span | | | 3.21 | 0.17 | 0.90 | 0.24 | 0.94 |
| MLM | Yes | | 3.38 | 0.18 | 0.90 | 0.24 | 0.91 |
| Prefix | | | 3.25 | 0.14 | 0.90 | 0.34 | 1.00 |

Table 6: Evaluation of DDLM with different masking strategies, $t_{\max}$ values, and with/without time warping for Enclosed-32 generation setting. The metrics with values $< 0.5$ (indicating highly repetitive samples) are displayed in colored font. We bolded the best metric values across other runs. See Section B for more details.

| L | H | D | Seq. len. | Masking | Optim. | Time Warping |
|---|---|---|---|---|---|---|
| 8 | 8 | 1024 | 64 | [MLM, Prefix, Span] | Adam | [no, yes] |
| LR | Scheduler | Warmup | Batch size | $t_{max}$ | Steps | |
| 3e-5 | Cos. w/ Warmup | 10k | 1024 | [10, 50, 300] | 1e6 | |

Table 7: Pre-training hyperparameters used for experiments with noise scheduling (See Section B). L stands for number of layers, H for number of heads in the Transformer layer, and D for hidden size.

