# OpenReview forum: "Democratized Diffusion Language Model"
_ICLR.cc/2024/Conference — Submitted to ICLR 2024_

### Official Review · Reviewer_UKex · 2023-10-29

**Soundness:** 2 fair
**Presentation:** 2 fair
**Contribution:** 2 fair
**Rating:** 3
**Confidence:** 3

**Summary:**

The paper studies Diffusion Models for text generation. It exams the runtime distinction between different diffusion models such as SSD, Plaid and CDCD.

A key observation from the research is the ability of most models to halt the generation process, enabling a faster text generation termed as "adaptive early exit" without diminishing the quality of the output.

The author also shares an open source re-implementation of the Diffusion LM trained with CDCD framework.

**Strengths:**

The paper studies a new field for text generation using diffusion models.  Given the inherent complexities and resource-intensive nature of running diffusion models continuously during generation, the research investigates the feasibility of early exiting by monitoring token switches across various pre-training checkpoints. The methodology of evaluating Cos between the score function and L2 norm the sample embeddings, and subsequently observing score angle changes, provides a novel insights to assess diffusion models.

**Weaknesses:**

The paper focuses on the concept of early stopping in diffusion models, which is an idea that has been previously explored, as noted in "Accelerating Diffusion Models via Early Stop of the Diffusion Process" as an example. The contribution to extend to text generation needs to be assessed. The technique of early stopping is a recognized practice during the inference stage of diffusion models. While the current paper's examination of token switches across different pre-training checkpoints offers a fresh angle, the approach's broader implications and significance in comparison to established methodologies could be further elucidated.

From Table 1 main results, we can see the choice of steps also provides very marginal impact to the final performance. It might be beneficial for the research to delve deeper into how this method stands out from or builds upon existing techniques in the field of diffusion models.

**Questions:**

LLM as Judge has some known weakness, from the provided prompts [Judging LLM-as-a-Judge with MT-Bench and Chatbot Arena], it seems the prompt is very simple and does not consider LLM bias. Can author comment on this?

---

> ### Author Response · Authors · 2023-11-17
> **Ukex answer**
>
> Thank you for your review. While performing an early exit is a well-established approach in Deep Learning, performing it for text diffusion was not previously explored. Furthermore, since the generation is performed in categorical space, we were able to study novel exit methods for diffusion early exiting.
>
> - From Table 1 main results, we can see the choice of steps also provides very marginal impact to the final performance. It might be beneficial for the research to delve deeper into how this method stands out from or builds upon existing techniques in the field of diffusion models.
>
> This is not actually correct. As seen with the Unconditional generation setup, reducing the total number of steps could lead to a 0.2-0.3 drop in the AR-NLL metric, which is not marginal for fixed dist-N values. Furthermore, based on our experiments, early exiting could be used in two scenarios – when we want to achieve a better quality of samples with a given number of generation steps or when we want to reduce the number of generation steps for a given quality of samples. Thus studying number of generation steps is orthogonal to early exiting methods.
>
> With our paper, we aimed to understand the behavior of DLMs, which is different from continuous diffusion models. We established new sampling methods and provided insights into reasons for such behavior. Understanding this behavior makes it possible to develop connections to other models and approaches, but without precisely studied groundings, it is impossible.
>
> - LLM as Judge has some known weakness, from the provided prompts [Judging LLM-as-a-Judge with MT-Bench and Chatbot Arena], it seems the prompt is very simple and does not consider LLM bias. Can author comment on this?
>
> Our task is significantly easier than usually accessed with LLM as a Judge. We do not want to select the best answer across a set of responses (e.g., alignment side-by-side setup). Instead, we used LLM to compare a generation from a specific step with the final generation (grammar, similar words, etc.). We observed that LLM provided a valuable estimation of such differences.

---

> > ### Comment · Reviewer_UKex · 2023-11-21
> >
> > Thank you for the clarification. My feedback aligns with other reviewers regarding the use of GPT-4 for evaluating relative scores, which requires further validation.
> >
> > The author should provide more empirical study to support "Our task is significantly easier than usually accessed with LLM as a Judge. We do not want to select the best answer across a set of responses (e.g., alignment side-by-side setup). Instead, we used LLM to compare a generation from a specific step with the final generation (grammar, similar words, etc.). We observed that LLM provided a valuable estimation of such differences."
> >
> > The current "GPT-SCORE DETAILS
> > The instruction contained a request to evaluate a text’s spelling, consistency, and coherence with a
> > number from 1 to 10 compared to the sampling from the last 1000-th generation step, which served
> > as a reference. Also, we included requesting for ignoring abrupt endings of texts since all models
> > were evaluated with sample length equal to 64." is questionable in terms of whether LLM will likely produce judgement bias towards certain number (e.g. smaller or larger numbers)

---

> > > ### Author Response · Authors · 2023-11-21
> > >
> > > We have made an important update to our paper based on your feedback. After careful consideration, we have decided to remove the controversial experiments involving GPT-4 evaluation from our study. We understand and respect your concerns regarding the applicability of GPT-4 in our analysis.
> > >
> > > However, we want to emphasize that we have retained the WER side-by-side evaluation, which still provides valuable insights into the convergence of the sampling algorithms. We believe that this evaluation metric remains reliable and informative for assessing the performance of our proposed approach.
> > >
> > > We kindly request you to evaluate the latest revision of our paper, which reflects the removal of the controversial experiments and addresses the issues and suggestions raised in your reviews. Your feedback has been invaluable in guiding the improvements we have made.
> > >
> > > Thank you for your time and consideration.

---

### Official Review · Reviewer_Sk8v · 2023-11-01

**Soundness:** 3 good
**Presentation:** 2 fair
**Contribution:** 2 fair
**Rating:** 3
**Confidence:** 4

**Summary:**

This paper reimplemented a Diffusion LM (DLM) trained with the CDCD framework and provided some analysis of DLMs. Besides, the paper showed that the generation process of most DLMs for general text generation can be halted.

**Strengths:**

1. This paper reimplemented the CDCD framework. If the code and checkpoint can be open-sourced, it can provide support for the research of DLMs.

2. This paper makes sufficient experiments and analysis on the existing DLMs, and obtains the early stopping strategy of DLMs by observing the AR-NLL curve.

**Weaknesses:**

1. The innovation of the paper is insufficient. The main contribution is to reproduce the CDCD structure and analyze the existing DLMs, without proposing new models or methods.

2. The length of the trained model is limited to 64, and it is not clear whether there will be different conclusions for longer lengths. The length of 64 is still a bit far from actual application.

3. We still care about the performance of pre-trained models on downstream tasks, and the paper did not select some downstream tasks for evaluation.

4. Writing issues:

    (1) The main contribution of the paper, such as the analysis of DLMs, is not given in the title. The writing style is a bit messy.

    (2) It is recommended to add the model parameter quantity to the comparison in Table 1.

**Questions:**

Does the model have the ability to output the </s> token? The sentences in Appendix D are all truncated results.

---

> ### Author Response · Authors · 2023-11-17
> **Sk8v answer**
>
> Thank you for your review!
>
> - The innovation of the paper is insufficient. The main contribution is to reproduce the CDCD structure and analyze the existing DLMs, without proposing new models or methods.
>
> The main aim of our paper is to understand the generation process of DLMs. With novel methodology in this direction, we observed that existing DLMs allow performing early exiting. We proposed several methods for performing early exiting (See Sections 4.2, 4.3). This work proposes faster sampling algorithms for DLMs and also lightens future research directions to further understanding and analysis of DLMs, which could lead to more new methods.
>
> - The length of the trained model is limited to 64, and it is not clear whether there will be different conclusions for longer lengths. The length of 64 is still a bit far from actual application.
>
> We will add experiments with longer sequences to the rebuttal revision soon. Please consider these new experiments when reevaluating your score.
>
> - We still care about the performance of pre-trained models on downstream tasks, and the paper did not select some downstream tasks for evaluation.
>
> Please note that we do not propose new DLMs with our paper. Instead, we studied sampling algorithms for existing models. These sampling algorithms are not connected to downstream task performance.
>
> - (1) The main contribution of the paper, such as the analysis of DLMs, is not given in the title. The writing style is a bit messy.
>
> We desired to highlight our trained DDLM model. Also, we wanted to have a connection with early exiting algorithms that allow faster sampling, thus making DLMs closer to practical usage by other people
>
> - (2) It is recommended to add the model parameter quantity to the comparison in Table 1.
>
> Thank you for this proposal; we have added the number of model parameters in the latest revision.
>
> - Does the model have the ability to output the </s> token? The sentences in Appendix D are all truncated results.
>
> Sentences in the Appendix D section were sampled with a sequence length of 64. Thus, they are truncated. All used in our experiments model could produce </s> tokens. For example, it is possible to set it as a condition at any specific point within the sequence, or the model itself could generate it for long enough sequences.

---

> ### Comment · Reviewer_Sk8v · 2023-11-22
>
> Thank you for the clarification. But my concerns are merely addressed.
>
> (1) I still harbor doubts regarding the accuracy of the title, and the innovativeness of the paper may not meet the standards set by ICLR.
>
> (2) Despite the authors' claim of not proposing a new DLM, the downstream tasks remain a subject worthy of investigation.
>
> (3) Experiments with longer text have yet to be provided.

---

> > ### Author Response · Authors · 2023-11-22
> >
> > Please note that experiments with longer sequences are already added to the latest revision

---

> > > ### Comment · Reviewer_Sk8v · 2023-11-22
> > >
> > > If I haven't missed anything, experiments with longer textual content are only displayed in Figure 8. However, the paper's most crucial DDLM lacks experimental results. Additionally, there are no actual text generation results presented, including longer texts and results with </s>.

---

### Official Review · Reviewer_E1db · 2023-11-02

**Soundness:** 2 fair
**Presentation:** 3 good
**Contribution:** 3 good
**Rating:** 5
**Confidence:** 5

**Summary:**

This paper explores the use of diffusion models for text generation and compares different frameworks used in the process (CDCD, Plaid, SSD). The authors focus on the sampling process and propose an adaptive early exit mechanism to accelerate text generation without compromising quality. The main contributions:
- re-implementation of the diffusion language model trained with the CDCD framework.
- propose and evaluate three adaptive criteria for early exiting
- side-by-side assessment to show the convergence of generation

**Strengths:**

- Re-implementation benifts the community.
- The step-by-step analysis could help us understand the generation process of diffusion models.

**Weaknesses:**

- It is good to see analysis of sampling between different diffusion models, however, no further explanation about the deep reason to cause these differences.
- The advantage of DDLM is to early exit and speedup the generation process. However, compared with some faster ODE solvers (e.g. DPM-solver[1]), early exit of DDLM maybe not superior than them.
- Early exit leads to the downgrade of generation diversity.

**Questions:**

- Why after your findings that using a noise scale of 0.9 is optimal, you still use a scale of 1.0 in later experiments
- In table 2, it is weird AR-NLL=0.44 and dist_1=0 with noise=0, do you have generation examples?
- The GPT-Score is a relative value, with step-1000 as the reference text. However, the reference text may not be fluent. Is it possible to obtain the absolute value?
- Can you also compare with discrete diffusion models?
- Why in Fig5, the NLL of Plaid (c) (<3.6) is lower than the DDLM (a) (>3.68)? This contradicts to the main table.

---

> ### Author Response · Authors · 2023-11-17
> **Answer E1db**
>
> Thank you for your review!
>
> - It is good to see analysis of sampling between different diffusion models, however, no further explanation about the deep reason to cause these differences.
>
> Differences between different diffusion models are specific to different training objectives and could be caused by noise scheduling. Experiments with varying noise scales ground this hypothesis. However, we did not dig into this and focused on proposing and understanding sampling methods for different DLMs and establishing an experimental setup for such research direction. Understanding various ways to perform an early exit for various models first could provide more evidence and grounding for further research on this topic rather than making solid assumptions without proper evaluation.
>
> - The advantage of DDLM is to early exit and speedup the generation process. However, compared with some faster ODE solvers (e.g. DPM-solver[1]), early exit of DDLM maybe not superior than them.
>
> First, let us clarify that DPM-solver is an second-order method, which implies two model’s forward passes during one step. Also note that DPM-solver is very similar to Heun sampler from [1]. In our experiments this type of samplers do not outperform Euler sampler with similar number of forward passes.
>
> [1] https://arxiv.org/abs/2206.00364
>
> - Early exit leads to the downgrade of generation diversity.
>
> We did not observe such behavior during our experiments. We hypothesize that you refer to Table 2 with experiments on noise scaling to allow faster convergence of the sampling algorithm. Lower diversity is expected since trimming the noise scale leads to “more deterministic” sampling. We will add plots with sampling diversity akin to Figure 5 in Section 4.3 to prove that early exiting does not reduce samples diversity in the rebuttal revision. Please refer to it for this weak point when reevaluating your score.
>
> - Why after your findings that using a noise scale of 0.9 is optimal, you still use a scale of 1.0 in later experiments
>
> The purpose of experiments with noise scaling is to understand the underlying behavior of DLMs, leading to emerging capabilities to perform an early exiting. However, we observed that performing generation with a slightly reduced noise scale without hurting AR-NLL and Dist values, such a setup is not conventional. In our case, this finding could provide insights on early exiting for DLMs for future works, but performing experiments in such configuration for other models is not conventional.
>
> - In table 2, it is weird AR-NLL=0.44 and dist_1=0 with noise=0, do you have generation examples?
>
> This is caused by AR-NLL metric issues since it is evaluated with external LM. It is conventional to obtain small values for repetitive samplings (e.g., one word repeated 64 times). Because of the issues with this metric, we also incorporated other metrics, such as a number of distinct tokens and self-bleu. E.g. Table 2 shows that the Dist value reached 0, indicating that AR-NLL does not provide helpful information.
>
> - The GPT-Score is a relative value, with step-1000 as the reference text. However, the reference text may not be fluent. Is it possible to obtain the absolute value?
>
> It is possible, although we observed that absolute value without side-by-side comparison with GPT-4 had a large variation. Performing a side-by-side comparison is established by other works (e.g., in Direct Preference Optimization), and to the best of our knowledge, no works used absolute evaluation with GPT-4 (doing so is an exciting research idea itself). In our case, we used it to understand the convergence of sampling algorithms. Even though the final sampling could not be fluent, this sampling is the one that is obtained with the original models as is. So, this experiment shows that early exiting samples after a specific step do not differ from the final sampling (see Appendix Section D).
>
> - Can you also compare with discrete diffusion models?
>
> We are unaware of discrete diffusion models capable of generating unconditional text. Existing unconditional general-purpose text generation models are continuous, so we performed experiments with them.
>
> - Why in Fig5, the NLL of Plaid (c) (<3.6) is lower than the DDLM (a) (>3.68)? This contradicts to the main table.
>
> For Figure 5, we used 1000 samples to evaluate the models, while the main table used 5000 samples. This led to a slight variation in AR-NLL values. We added an explanation to the latest revision.

---

> > ### Comment · Reviewer_E1db · 2023-11-21
> > **Thank you for clarification**
> >
> > Thank you for your clarification. Some of my questions are resolved. But I still maintain these concerns: (1) Early stop, as one of your main contribution, is saving ~30% steps according to Fig 5 and this improvement is limited. (2) Whether using GPT-4 to evaluate the relative score is responsible or not needs further discussion.

---

> > > ### Author Response · Authors · 2023-11-21
> > >
> > > We have made an important update to our paper based on your feedback. After careful consideration, we have decided to remove the controversial experiments involving GPT-4 evaluation from our study. We understand and respect your concerns regarding the applicability of GPT-4 in our analysis.
> > >
> > > However, we want to emphasize that we have retained the WER side-by-side evaluation, which still provides valuable insights into the convergence of the sampling algorithms. We believe that this evaluation metric remains reliable and informative for assessing the performance of our proposed approach.
> > >
> > > We kindly request you to evaluate the latest revision of our paper, which reflects the removal of the controversial experiments and addresses the issues and suggestions raised in your reviews. Your feedback has been invaluable in guiding the improvements we have made.
> > >
> > > Thank you for your time and consideration.

---

### Author Response · Authors · 2023-11-20
**General response on rebuttal revision**

Dear Reviewers,

We have uploaded a new revision of our paper. With this version, we

1. We have conducted additional experiments using longer sequences and demonstrated that the early exiting behavior remains consistent (see Figure 8 in the appendix). This analysis provides further evidence of the effectiveness and resilience of our proposed approach.

2. To address your concern regarding the variability of samples, we have included an evaluation of the number of unique tokens for different sampling methods (see Figure 7 in the appendix). This experiment clearly illustrates that early exiting does not compromise the variability of generated samples. This observation strengthens the validity and robustness of our proposed method.

3. Based on your feedback, we have diligently fixed various stylistic issues, including more accurate reporting of the number of model parameters and other minor details. These improvements ensure the accuracy and clarity of our paper.

We are available to address any further questions or concerns you may have. We kindly request that you consider revising your scores to reflect our improvements, considering the amendments that directly address the weaknesses you noted.

Once again, we extend our heartfelt gratitude for your valuable feedback and contribution to refining our research.

---

### Comment · Area_Chair_g3et · 2023-11-21
**response to authors**

Dear Reviewers,

Please kindly review the authors' rebuttal to assess if their revisions and responses have influenced your opinions.

Thank you,

AC

---

### Meta-Review · Area_Chair_g3et · 2023-12-08

**Metareview:**

The paper explores text generation using diffusion language models, comparing frameworks and suggesting an adaptive early exit mechanism. While strengths include a detailed analysis of diffusion language models, weaknesses involve a lack of novel methods, incomplete explanations for sampling differences, and potential drawbacks in diversity due to early exit. Concerns also arise from findings based on a model length of 64 and the absence of downstream task evaluation.

**Justification For Why Not Higher Score:**

All reviewers recommended rejections. Reasons to reject include insufficient novelty, limited exploration of longer model lengths, a lack of practical applicability assessment through downstream tasks, criticism of writing style and clarity, insufficient comparative analysis, and incomplete explanations.

**Justification For Why Not Lower Score:**

N/A

---

### Decision · Program_Chairs · 2024-01-16

Reject